# Functional Characterization of Transforming Growth Factor-β Signaling in Dasatinib Resistance and Pre-BCR^+^ Acute Lymphoblastic Leukemia

**DOI:** 10.3390/cancers15174328

**Published:** 2023-08-30

**Authors:** Gila Mostufi-Zadeh-Haghighi, Pia Veratti, Kyra Zodel, Gabriele Greve, Miguel Waterhouse, Robert Zeiser, Michael L. Cleary, Michael Lübbert, Jesús Duque-Afonso

**Affiliations:** 1Department of Medicine I, Medical Center—University of Freiburg, Faculty of Medicine, University of Freiburg, 79106 Freiburg, Germany; gila.mostufi@uniklinik-freiburg.de (G.M.-Z.-H.); pia.veratti@uniklinik-freiburg.de (P.V.); kyra.zodel@uniklinik-freiburg.de (K.Z.); gabriele.greve@uniklinik-freiburg.de (G.G.); miguel.waterhouse@uniklinik-freiburg.de (M.W.); robert.zeiser@uniklinik-freiburg.de (R.Z.); michael.luebbert@uniklinik-freiburg.de (M.L.); 2German Cancer Consortium (DKTK), Partner Site Freiburg, 79106 Freiburg, Germany; 3German Cancer Research Center (DKFZ), 69120 Heidelberg, Germany; 4Department of Pathology, Stanford University, Stanford, CA 94305, USA; mcleary@stanford.edu

**Keywords:** acute lymphoblastic leukemia, B-cell precursors, transforming growth factor-β signaling pathway, dasatinib, drug resistance

## Abstract

**Simple Summary:**

We focus on the characterization of the transforming growth factor-β (TGFβ) signaling pathway in B acute lymphoblastic leukemia (ALL) and in resistance to the multi-kinase inhibitor dasatinib in order to provide a better understanding of the molecular and functional mechanisms underlying leukemic transformation and the development of drug resistance. We provide evidence that TGFβ signaling is an important negative regulator of cell growth in B-cell precursor- ALL as well as in generated dasatinib-resistant ALL cells.

**Abstract:**

The multi-kinase inhibitor dasatinib has been implicated to be effective in pre-B-cell receptor (pre-BCR)-positive acute lymphoblastic leukemia (ALL) expressing the E2A-PBX1 fusion oncoprotein. The TGFβ signaling pathway is involved in a wide variety of cellular processes, including embryonic development and cell homeostasis, and it can have dual roles in cancer: suppressing tumor growth at early stages and mediating tumor progression at later stages. In this study, we identified the upregulation of the TGFβ signaling pathway in our previously generated human dasatinib-resistant pre-BCR^+^/E2A-PBX1^+^ ALL cells using global transcriptomic analysis. We confirm the upregulation of the TGFβ pathway member SMAD3 at the transcriptional and translational levels in dasatinib-resistant pre-BCR^+^/E2A-PBX1^+^ ALL cells. Hence, dasatinib blocks, at least partially, TGFβ-induced SMAD3 phosphorylation in several B-cell precursor (BCP) ALL cell lines as well as in dasatinib-resistant pre-BCR^+^/E2A-PBX1^+^ ALL cells. Activation of the TGFβ signaling pathway by TGF-β1 leads to growth inhibition by cell cycle arrest at the G0/G1 stage, increase in apoptosis and transcriptional changes of SMAD-targeted genes, e.g. c-MYC downregulation, in pre-BCR+/E2A-PBX1+ ALL cells. These results provide a better understanding about the role that the TGFβ signaling pathway plays in leukemogenesis of BCP-ALL as well as in secondary drug resistance to dasatinib.

## 1. Introduction

Acute lymphoblastic leukemia (ALL) represents the most common cancer in children and adolescents, accounting for 25% of all childhood malignancies, and derives in the majority of cases from B-cell precursors (BCP) [1]. Although five-year overall survival rates have increased to over 85% in the last 40 years through improved treatment regimens, relapse rates still range between 25 and 30% in certain risk groups [2,3,4,5]. Therefore, there is a clinical need for the development of novel treatment approaches. 

Several studies have suggested the multi-kinase inhibitor dasatinib—FDA-approved for treatment of chronic myeloid leukemia (CML) and Philadelphia (Ph) chromosome-positive ALL [6,7]—to be a promising targeted therapeutic agent in a distinct subtype of BCP-ALL, i.e., pre-B-cell receptor (pre-BCR)^+^/E2A-PBX1^+^ALL. Pre-BCR^+^ ALL occurs in 10–15% of BCP-ALL cases and is characterized by the expression of functional pre-BCR mediating survival during early B-cell development. Half of pre-BCR^+^ ALL cases are associated with the chromosomal translocation t(1;19) (q23;p13) coding for the oncogenic transcription factor E2A-PBX1 (TCF3-PBX1) [8,9,10,11,12,13,14,15]. However, the rapid development of drug resistance limiting the response to targeted therapies with small molecule inhibitors requires a deeper understanding of the underlying mechanisms in order to improve therapeutic efficacy and limit toxicity [16,17,18]. We previously generated a cell-culture based model of dasatinib resistance using the human pre-BCR^+^/E2A-PBX1^+^ ALL cell line RCH-ACV, which has been used to study mechanisms of resistance to targeted therapy in this genetic ALL subtype [12].

In the current study, transcriptomic analysis identified the transforming growth factor-β (TGFβ) signaling pathway to be significantly upregulated in dasatinib-resistant pre-BCR^+^/E2A-PBX1^+^ ALL cells, which was further investigated by functional characterization studies in pre-BCR^+^/E2A-PBX1^+^ as well as BCP-ALL cells. TGFβ signaling activity was found to be heterogeneous in BCP-ALL, which can be partially blocked by dasatinib. Moreover, we show that TGFβ signaling negatively regulated cell proliferation in BCP-ALL cell lines as well as in dasatinib-resistant pre-BCR^+^/E2A-PBX1^+^ RCH-ACV cells. 

## 2. Materials and Methods

### 2.1. Human Precursor B-ALL Cell Lines

Human pre-BCR^+^/E2A-PBX1^+^ ALL cell lines RCH-ACV (RRID: CVCL_1851) and 697 (RRID: CVCL_0079) and pre-BCR^−^/E2A-PBX1^−^ ALL cell lines REH (RRID: CVCL_1650), HAL-01 (RRID: CVCL_1242) and SEM (RRID: CVCL_0095) were obtained from DSMZ (Leibniz Institute, Braunschweig, Germany) and cultured in Gibco™ RPMI medium (Thermo Fisher Scientific, Waltham, MA, USA) supplemented with 10% fetal bovine serum, 100 U/mL penicillin, 100 µg/mL streptomycin and 2 mM L-Glutamine. The SEM cell line was cultured in Gibco™ IMDM medium (Thermo Fisher Scientific) supplemented with 10% fetal bovine serum, 100 U/mL penicillin, 100 µg/mL streptomycin, 4 mM L-Glutamine and 25 mM HEPES. Dasatinib-resistant cell lines were generated and described previously [12]. Two independent dasatinib-resistant clones were generated in parallel after the incubation of the pre-BCR^+^/E2A-PBX1^+^ RCH-ACV cell line with increasing concentrations of dasatinib for at least 9 months. Global transcriptomic profiles were generated to study differences between both clones in previous studies [12]. All human cell lines have been authenticated using STR (or SNP) profiling within the last three years. All experiments were performed with mycoplasma-free cells. 

### 2.2. Gene Expression Analysis

RNA was prepared using a Nucleo Spin RNA-Kit (MACHERY-NAGEL, Düren, Germany). RNA was reverse transcribed to single-stranded cDNA with SuperScriptIII Reverse Transcriptase and oligo (dT)_20_ 50 µM (Thermo Fisher Scientific). Primer pairs for *GAPDH*, *SMAD3* and c*-MYC* were obtained from Apara-bioscience GmbH (Denzlingen, Germany), *p15Ink4b* from Sigma-Aldrich Inc. (St. Louis, MO, USA) and *p21Cip1* from Integrated DNA Technologies IDT, Inc. (Coralville, IA, USA). Primer sequences are listed in Appendix A. Gene expression was analyzed using the LightCycler^®^ 480 RT-qPCR and LightCycler^®^ 480 SYBR Green I Master (Roche, Basel, Switzerland). Relative quantification of gene expression was performed as indicated. RNA sequencing and gene set enrichment analysis of generated dasatinib-resistant cells were previously described [12]. (Gene Expression Omnibus—http://ncbi.nlm.nih.gov/geo accession number GSE97352, accessed on 3 April 2017.

### 2.3. Western Blot Analysis

Validation of SMAD3 protein expression was performed by Western blot analysis using the NuPAGE^®^ system (Thermo Fisher Scientific) according to manufacturer’s protocol. Whole-cell extract was prepared using a Nuclear Extract Kit (Active Motif, Carlsbad, CA, USA), and protein concentration was analyzed using the BCA Protein Assay Kit (Thermo Fisher Scientific). Protein lysates were subjected to denaturing SDS gel-electrophoresis using NuPAGE^®^ Novex^®^ Bis-Tris Pre-Cast Gels 4–12%, and protein transfer was performed on Immobilon-P PVDF membrane 0.45 µm (Millipore—Merck, Darmstadt, Germany). Membranes were incubated overnight at 4 °C with the following primary antibodies: anti-SMAD3 Rabbit Antibody (#2118S) and anti-GAPDH (14C10) rabbit monoclonal antibody (#2118S). Subsequently, membranes were incubated for 1 h at room temperature with secondary HRP-linked antibody against rabbit-IgG (#7074P2). Antibodies were obtained from Cell Signaling Technology Inc. (Danvers, MA, USA). Proteins were detected by enhanced chemoiluminescence using SuperSignal^®^ West Pico and Femto (Thermo Scientific) and an ECL CHEMOCAM IMAGER (INTAS Science Imaging Instruments GmbH, Göttingen, Germany). Protein quantification by densitometry was assessed using ImageJ64 Software 1.47v, Java 1.8.0_112 (64-bit) (NIH, Bethesda, MD, USA).

### 2.4. Phospho-Specific Intracellular Flow Cytometry Assay

Dasatinib-sensitive and -resistant preBCR^+^/E2A-PBX1^+^ RCH-ACV clones as well as BCP-ALL cell lines were seeded at 1 × 10^6^ cells/mL and incubated for 1 h at 37 °C and 5% CO_2_. Cells were treated with the vehicle (DMSO or H_2_O+BSA 0.1%), dasatinib or TGF-β1 alone and in combination at the indicated concentrations for 30 min, respectively. Cells were fixed with 1.5% formaldehyde for 10 min at room temperature and permeabilized with 1 mL ice-cold methanol for 20 min on ice [19]. 

Staining was conducted with BD Alexa Fluor^®^ 647 anti-SMAD2(pS465/pS467)/SMAD3(pS423/pS425) antibody (BD Biosciences, San Jose, CA, USA). Assessment of phosphorylation levels of SMAD2/3 proteins was performed on flow cytometry BD LSRFortessa™ using BD FACS DIVA Software, v9.0.1 (BD Biosciences). Data analysis was performed using Flow Jo Data Analysis Software, v10.8.1 (BD, Franklin Lakes, NJ, USA). Median Fluorescence Intensity (MFI) values of treated samples were normalized to vehicle-treated samples, with results shown as the mean of three independent experiments.

### 2.5. Cell Proliferation Assays

Cells were seeded at 2.5 × 10^5^ per mL of fresh medium supplemented with the compounds TGF-β1 (Pepro Tech Inc., Rocky Hill, NJ, USA) or dasatinib (Selleckchem, Houston, TX, USA), as indicated, for 6 or 12 days. Cell proliferation and viability was assessed every 3 days by trypan blue exclusion assay using Gibco™ Trypan Blue Stain (0.4%) (Thermo Fisher Scientific), cells were split to initial concentration, re-suspended in fresh medium and treated with the indicated concentrations. Data are expressed as cell proliferation relative to vehicle, calculated from accumulative data over the respective days of treatment.

### 2.6. Cell Cycle and Apoptosis Analysis

Dasatinib-sensitive and -resistant RCH-ACV clones were subjected to in vitro long-term drug treatment as described above. The concentration of TGF-β1 was optimized to detect an effect in cell cycle and apoptosis analysis. Cell proliferation was markedly reduced at concentrations of 0.5 and 1.0 ng/mL; however, correlating changes in cell cycle or apoptosis assays were detected at 5.0 and 10.0 ng/mL by flow cytometry. Further experiments were performed with TGF-β1 5.0 ng/mL. After 6 days of treatment, cell cycle and apoptosis/cell death were analyzed by flow cytometry on BD LSRFortessa™ using BD FACS DIVA Software, v9.0.1 (BD Biosciences). Following 1 h of BrdU incorporation, cell cycle was assessed using BD Pharmingen™ FITC BrdU Flow Kit (BD Biosciences); and apoptosis/cell death was detected using the BD Pharmingen™ FITC Annexin V Apoptosis Detection Kit I (BD Biosciences) according to manufacturer’s instructions. Data analysis was performed using Flow Jo Data Analysis Software, v10.8.1 (BD). 

### 2.7. Statistical Analysis

To analyze statistically significant differences between data sets, one-sample and unpaired two-tailed Student’s *t*-tests were performed, unless otherwise stated. Data represent the mean of three independent experiments, with error bars indicating ± standard deviation (SD). Differences between mean were considered significant at *p* < 0.05 (*); *p* < 0.01 (**); *p* < 0.001 (***); or not significant (n.s.). Statistical analysis was performed using GraphPad Prism 5.03 Software (San Diego, CA, USA).

## 3. Results

### 3.1. Upregulation of the TGFβ Signaling Pathway in Acquired Dasatinib Resistance

Using gene set enrichment analysis (GSEA) of the Kyoto Encyclopedia of Genes and Genomes (KEGG) pathways, we identified the TGFβ signaling pathway to be significantly enriched in the previously generated dasatinib-resistant pre-BCR^+^/E2A-PBX1^+^ RCH-ACV cell line (Figure 1a,b) [12]. Gene expression analysis by RNA-seq revealed several members of the TGFβ superfamily to be upregulated in dasatinib-resistant RCH-ACV cells (Figure 1c,d), among which were the TGFβ subfamily members *TGFβ2* and *SMAD3* as well as different bone morphogenetic protein (BMP) subfamily members, such as *BMPR1A*, *BMP2* and *BMP8B*. Although SMAD3 was initially identified to be upregulated in dasatinib-resistant RCH-ACV cells from clone 2 by RNA sequencing, validation experiments by RT-qPCR and Western blot confirmed significant upregulation in dasatinib-resistant RCH-ACV cells in both clones (Figure 1e,f). We also examined the mRNA levels of other TGFβ superfamily members found to be upregulated by RNAseq and confirmed higher expression of *BMP2* in dasatinib-resistant RCH-ACV clones (Appendix A). Both the TGFβ and BMP signaling pathways are involved in regulating a wide variety of different functions in adult tissues as well as during embryonic development [20,21]. The TGFβ/SMAD signaling pathway is a crucial regulator of proliferation in many different cell types and is also implicated in cancer [22]. We therefore continued further functional studies with SMAD3.

### 3.2. Dasatinib Partially Inhibits TGF-β1-Induced SMAD2/3 Phosphorylation in Pre-BCR^+^/E2A-PBX1^+^ ALL Cells

Phosphorylated SMAD proteins are central effectors of TGFβ signaling as they act both as signal transducers downstream of a hetero-tetrameric receptor complex, consisting of type I and II receptors, and as transcriptional regulators at the nuclear level [23,24]. To functionally understand the mechanism of action of dasatinib on TGFβ signaling activity in dasatinib-sensitive and -resistant RCH-ACV cells, we examined the phosphorylation state of SMAD2/3 proteins by intracytoplasmic flow cytometry following treatment with dasatinib and/or TGF-β1. SMAD2/3 phosphorylation was induced in a dose-dependent manner by TGF-β1 and partially inhibited by dasatinib in both dasatinib-sensitive and -resistant RCH-ACV cells (Figure 2). No difference in SMAD2/3 phosphorylation levels was observed when comparing dasatinib-sensitive and -resistant cells. In clone 2, a decreasing trend of SMAD2/3 phosphorylation by dasatinib was observed when a high amount of TGF-β1 was used (Figure 2). 

### 3.3. Heterogeneous TGFβ Signaling Activity and Effects in Human BCP-ALL Cell Lines

In order to validate our findings and to identify possible underlying molecular features, we used several BCP-ALL cell lines with different cytogenetic backgrounds for studying the effects of dasatinib and TGF-β1 on TGFβ signaling activity in BCP-ALL. According to our previous studies, the E2A-PBX1^+^ 697 cells were moderately sensitive to dasatinib (IC50 100 nM–1 μM). In contrast, the E2A-HLF^+^ HAL-01, the MLL-AF4^+^ SEM and the TEL-AML1^+^ REH cell lines were resistant to dasatinib (IC50 > 1 μM) [12]. Using intracytoplasmic flow cytometry analysis of SMAD2/3 protein phosphorylation in different BCP-ALL cell lines, we found heterogeneous phosphorylated SMAD2/3 levels at the basal state following dose-dependent stimulation with TGF-β1 as well as following inhibition by dasatinib, which varied from partial to lacking responsiveness (Figure 3a,b).

Intermediate basal phosphorylation states were observed in the pre-BCR^+^/E2A-PBX1^+^ cell lines 697, in both dasatinib-sensitive and -resistant RCH-ACV, and moreover in the pre-BCR^−^/E2A-PBX1^−^ REH cell line, which all displayed an intermediate increase in TGF-β1-induced phosphorylation levels, which could be partially blocked by dasatinib (Figure 3a,b). In contrast, the pre-BCR^−^/E2A-PBX1^−^ cell line HAL-01 presented the lowest basal SMAD2/3 phosphorylation level while being the most sensitive to stimulation by TGF-β1, whereas the SEM cell line exhibited the highest basal phosphorylation level while showing only a slight increase in SMAD2/3 protein phosphorylation levels in response to TGF-β1 (Figure 3a,b). However, TGF-β1-induced phosphorylation of SMAD2/3 proteins in both cell lines was not significantly inhibited by dasatinib.

To investigate the functional implication of TGF-β signaling in B-ALL leukemogenesis, the BCP-ALL cell lines were subjected to treatment with TGF-β1 at different concentrations. We found that cell proliferation of all examined human BCP-ALL cell lines was at least partially inhibited by TGF-β1 stimulation (Figure 4). Interestingly, the E2A translocated cell lines RCH-ACV and 697 (E2A-PBX1^+^, respectively) as well as HAL-01 (E2A-HLF^+^) showed the highest sensitivity to TGF-β1 growth inhibitory effects compared to SEM (MLL-AF4^+^) and REH (TEL-AML1^+^). Our results indicate that cell lines arising from BCP-ALL retain at least partial responsiveness to TGF-β1 growth-inhibitory effects (Figure 4).

### 3.4. Cell Proliferation of Dasatinib-Resistant RCH-ACV Cells Inhibited by TGF-β1

To evaluate the anti-proliferative effects of TGF-β1 in dasatinib-resistant RCH-ACV cells and to further understand the underlying mechanisms, we exposed dasatinib-sensitive and -resistant RCH-ACV cells to treatment with dasatinib or TGF-β1 with subsequent flow cytometry analysis of cell cycle and apoptosis. Interestingly, we found that treatment with TGF-β1 also resulted in inhibition of cell proliferation in both dasatinib-sensitive and -resistant RCH-ACV clones (Figure 5a, Appendix A).

Hence, subsequent cell cycle and apoptosis analysis revealed similar TGF-β1-induced effects in both clones of dasatinib-sensitive and -resistant RCH-ACV cells (Figure 5b,c, Appendix A).

Cytostatic effects mediated by TGF-β1 have been linked to cell cycle arrest at the G0/G1 stage via transcriptional downregulation of the proto-oncogene c*-MYC* and induction of the cyclin-dependent kinase (CDK) inhibitors *p15Ink4b* (*CDKN2B*) and *p21Cip1* (*CDKN1A*) [25]. Target gene expression of genes involved in TGF-β1-induced cell cycle arrest was analyzed using RT-qPCR following treatment with TGF-β1 in dasatinib-sensitive and -resistant RCH-ACV cells. As expected, c*-MYC* expression was downregulated by TGF-β1 in dasatinib-resistant and -sensitive RCH-ACV clone 1 cells. Consistent with this, we observed TGF-β1-induced transcriptional activation of *p15Ink4b*. Unexpectedly, *p21Cip1* expression was downregulated following TGF-β1 treatment in dasatinib-sensitive but not in -resistant RCH-ACV cells (Figure 6). These data suggest *c-MYC* and *p15Ink4b* as possible drivers of the observed TGF-β1-induced cell growth arrest in pre-BCR+/E2A-PBX1+ ALL cells.

## 4. Discussion

In previous studies, we identified several enriched and depleted KEGG signaling pathways in a model of secondary dasatinib resistance in a distinct subtype of BCP-ALL, i.e., pre-BCR^+^/E2A-PBX1^+^, using gene ontology (GO) analysis [12]. In the present study, we employed a different approach using GSEA, identifying the TGFβ signaling pathway to be enriched by dasatinib-resistant pre-BCR^+^/E2A-PBX1^+^ ALL cells. Different members of the TGFβ and BMP subfamilies were found to be upregulated; however, knowing their role in cancer and cell proliferation regulation, we pursued further functional studies investigating the role of the TGFβ/SMAD signaling pathway in the preclinical model of dasatinib-resistance in pre-BCR^+^/E2A-PBX1^+^ ALL, as well as in several BCP-ALL cell lines with different cytogenetics. We cannot exclude the involvement of BMP signaling in contributing to the resistant phenotype of RCH-ACV cells to dasatinib, which requires additional investigation in future research.

### 4.1. Heterogeneous TGFβ Signaling Activity and TGF-β1-Induced Growth Inhibition Imply Functionally Distinct BCP-ALL Subsets

Our functional studies involving SMAD2/3 phosphorylation analysis and cell proliferation assays suggest that the examined BCP-ALL cell lines retain at least partial responsiveness to TGF-β1 cytostatic effects. However, they exhibit heterogeneous basal as well as TGF-β1-induced SMAD2/3 phosphorylation levels, which might have an impact on the TGF-β1-mediated anti-proliferative response. Among the examined cell lines, E2A-HLF^+^ HAL-01 cells showed the highest sensitivity to TGF-β1 in terms of induction of SMAD2/3 phosphorylation as well as cytostatic effects. In contrast, the MLL-AF4^+^ SEM cell line presented the lowest increase in SMAD2/3 phosphorylation levels following stimulation with TGF-β1. Moreover, they were the least sensitive to TGF-β1 anti-proliferative effects. Conversely, the pre-BCR^+^/E2A-PBX1^+^ cell lines RCH-ACV and 697 as well as the TEL-AML1^+^ REH cells displayed significant responsiveness to growth-inhibitory effects mediated by TGF-β1. Buske et al. reported high inter-individual heterogeneity in anti-proliferative effects within primary blasts from patients with common B-cell ALL treated in vitro with TGF-β1; however differences in cytogenetic background or possible underlying molecular features were not addressed in the study [26]. 

As previously described [12], dasatinib-resistant RCH-ACV cell clones grow much slower than dasatinib-sensitive clones (Figure 1a.). We hypothesize that the upregulation of the TGFβ signaling pathway might be implicated in the lower proliferation rate of dasatinib-resistant pre-BCR^+^/E2A-PBX1^+^ ALL cells, retaining its growth-inhibitory function. This is supported by a proliferation decrease observed after the treatment with TGF-β1 in several B-ALL cell lines with different karyotypes, including the TEL-AML1^+^ REH ALL cell line. In contrast, Ford et al. described previously that TGF-β1 did not further reduce cell proliferation of the murine pro-B cell line, Ba/F3, transduced with TEL-AML1 [27]. Neither dasatinib-sensitive nor -resistant pre-BCR^+^/E2A-PBX1^+^ RCH-ACV ALL cell lines show growth difference in response to TGF-β1. BCP-ALLs respond heterogeneously to TGF-β1, as shown by cell proliferation assays and pSMAD2/3 phosphorylation studies. The mechanisms that determine differences in growth inhibition in dasatinib-sensitive or -resistant cell lines and in BCP-ALL with different karyotypes in response to TGF-β1 were not systematically investigated in the study. Further studies are needed to elucidate the role of the TGFβ signaling pathway in dasatinib resistance.

### 4.2. Dasatinib Blocks TGF-β1-Induced SMAD2/3 Phosphorylation by Interacting with TGFβ Type I Receptors

We showed that signaling activation via phosphorylation of SMAD2/3 proteins in response to TGF-β1 is blocked in a heterogeneous manner by dasatinib in different BCP-ALL cell lines, as well as in the dasatinib-resistant RCH-ACV cells, and that inhibition of SMAD2/3 phosphorylation does not correlate with sensitivity to dasatinib [12]. Dasatinib was developed to interact mainly with the ABL and SRC families of tyrosine kinases, thus blocking intracellular signaling transduction in Ph^+^ CML and ALL [28,29]. Indeed, a study using drug affinity chromatography revealed that dasatinib targets more than 40 different kinases, among which were the TβRI, which present serine/threonine kinase activity [30], which was confirmed by further investigation using docking studies showing that dasatinib was able to bind the TβRI [31]; however, contradictory findings regarding its effect on SMAD protein activation have been previously reported, in which dasatinib inhibited TGF-β1-induced SMAD2/3 phosphorylation in pancreatic adenocarcinoma cells while inducing phosphorylation of SMAD3 after combined treatment with TGF-β1 and dasatinib in lung cancer cells [31,32]. Our studies provide further insight into the underlying features of inhibition of SMAD2/3 phosphorylation by dasatinib, which is dependent on the cell type and the basal phosphorylation state of the cell as well as on the TGF-β1-induced SMAD2/3 phosphorylation level. The functional implication of combined treatment with dasatinib and TGFβ in ALL cell lines should be investigated in future studies.

### 4.3. Differential Mechanisms Underlying TGF-β1 Growth-Inhibitory Effects in Dasatinib-Sensitive and -Resistant RCH-ACV Cells

Cell cycle analysis by flow cytometry revealed growth inhibition by TGF-β1 via induction of G0/G1 cell cycle arrest and increase in apoptotic subG0 cell fraction. However, target gene expression studies showed downmodulation of c*-MYC* and upregulation of the CDK inhibitor *p15Ink4b* in both dasatinib-sensitive and -resistant RCH-ACV cells. Inhibition of cell proliferation mediated by TGF-β1 has been mainly related to cell cycle arrest at the G0/G1 stage, whereas mechanisms regulating programmed cell death in response to TGF-β1 remain largely unknown [25]. Differential kinetics in the regulation of TGFβ signaling activity as well as of target gene expression have been previously described [20,22,33,34]. Moreover, TGFβ transcriptional programs and resulting biological functions are highly cell-type- and context-dependent, which might become impaired in malignant cells by aberrant or failed expression as well as loss of expression of TGFβ target genes and transcriptional co-factors involved in cell cycle regulation [20,22,25]. Furthermore, we cannot exclude the involvement of other signaling pathways counteracting the regulation of target gene expression and induction of cell cycle arrest by TGF-β1. Pre-BCR^+^/E2A-PBX1^+^ ALL cells are dependent on hyperactivated PI3K/AKT signaling [8,9,35,36], which is a well-known opponent of TGFβ-mediated cell-cycle arrest by phosphorylating and inhibiting FOXO transcription factors, thus preventing their nuclear translocation and cooperation with SMAD transcription factors in the transcriptional activation of *p15Ink4b* and *p21Cip1* [36,37,38]. Therefore, we hypothesize that inhibition of the PI3K/AKT pathway in pre-BCR^+^/E2A-PBX1^+^ ALL cells might have additive effects to antiproliferation mediated by TGF-β1. Future studies highlighting the complex web of interactions between TGFβ and other signaling pathways are needed to gain further insight into these questions.

In addition, cellular senescence and autophagy as alternative mechanisms of inhibiting cell proliferation have been suggested to be induced by TGFβ signaling, but these have not been investigated in this study [39].

## 5. Conclusions

In this study, we show that TGFβ signaling plays an important role in the negative regulation of cell proliferation in BCP-ALL cell lines as well as in dasatinib-resistant E2A-PBX1^+^ RCH-ACV cells. Although dasatinib shows promising early preclinical effects for the treatment of pre-BCR^+^/E2A-PBX1^+^ ALL, its ability to target a broad spectrum of different kinases might contribute to the onset of secondary drug resistance. For almost two decades, the TGFβ signaling pathway has not been in the spotlight of hematological cancer research, and approaches to target this pathway in various diseases remain challenging. Therefore, further characterization of the complex regulatory actions by which TGFβ signaling mediates its functions during normal lymphopoiesis and in its malignant counterparts should be the focus of future research in order to improve our understanding of this signaling pathway, as well as to develop novel treatment approaches in BCP-ALL. 

## Figures and Tables

**Figure 1 cancers-15-04328-f001:**
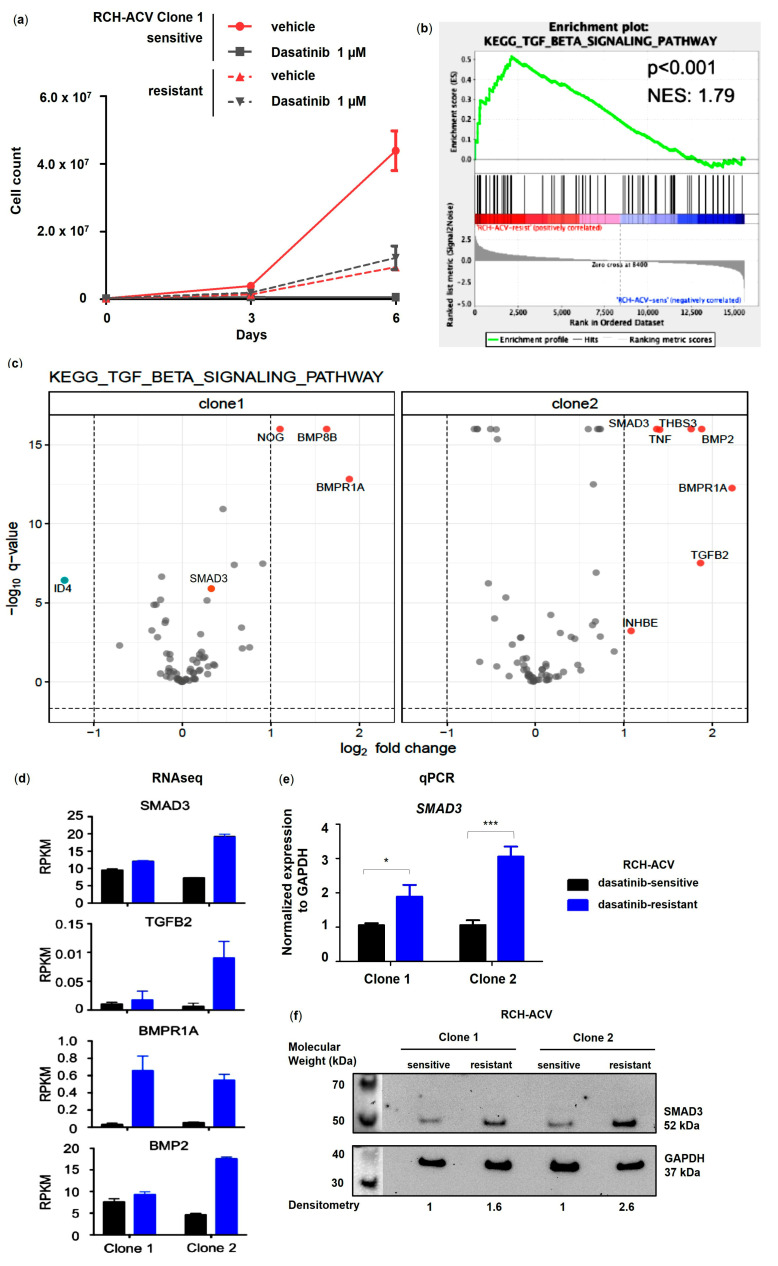
TGFβ pathway member SMAD3 is significantly upregulated in dasatinib-resistant E2A-PBX1^+^ ALL cells. (**a**) Long-term treatment in dasatinib-sensitive and -resistant RCH-ACV cell line after 6 days of treatment with vehicle or dasatinib. Cell number was assessed every 3 days by trypan blue exclusion assay. Cell proliferation is shown for RCH-ACV clone 1 as accumulative data and represents mean of three replicate experiments, with error bars indicating ± SD. (**b**) GSEA plot shows enrichment of TGFβ signaling pathway in dasatinib-resistant cells. (**c**) Volcano plots for differentially expressed genes (DEGs) in KEGG TGFβ signaling pathway for dasatinib-resistant compared to sensitive RCH-ACV clones. Red dots indicate upregulated genes; blue dots indicate downregulated genes. Q-values < 0.05 and log_2_ fold change <−1 and >1 were considered statistically significant. Expression data were visualized using R software, Version 4.0.3 (**d**) Reads per kilobase of transcript per million mapped reads (RPKM) values of selected genes involved in the TGFβ signaling pathway and upregulated in dasatinib-resistant RCH-ACV clones. Bar denotes mean and error bars, SEM (n = 3). (**e**) Gene expression of TGFβ pathway member *SMAD3* was validated using RT-qPCR. *GAPDH* was used as housekeeping gene. Expression of target gene is displayed as normalized to housekeeping gene mean concentration. Data are shown as mean of three replicate experiments, with error bars indicating ± SD. Statistical analysis by Student’s *t*-test (*: *p* < 0.05; ***: *p* < 0.001). (**f**) Western Blot analysis (representative of two independent experiments) shows upregulated protein levels in dasatinib-resistant RCH-ACV cells. Densitometry values were calculated using ImageJ software. Protein levels were normalized to GAPDH and expressed as fold change relative to dasatinib-sensitive RCH-ACV clone 1. Original western blots are presented in Appendix A.

**Figure 2 cancers-15-04328-f002:**
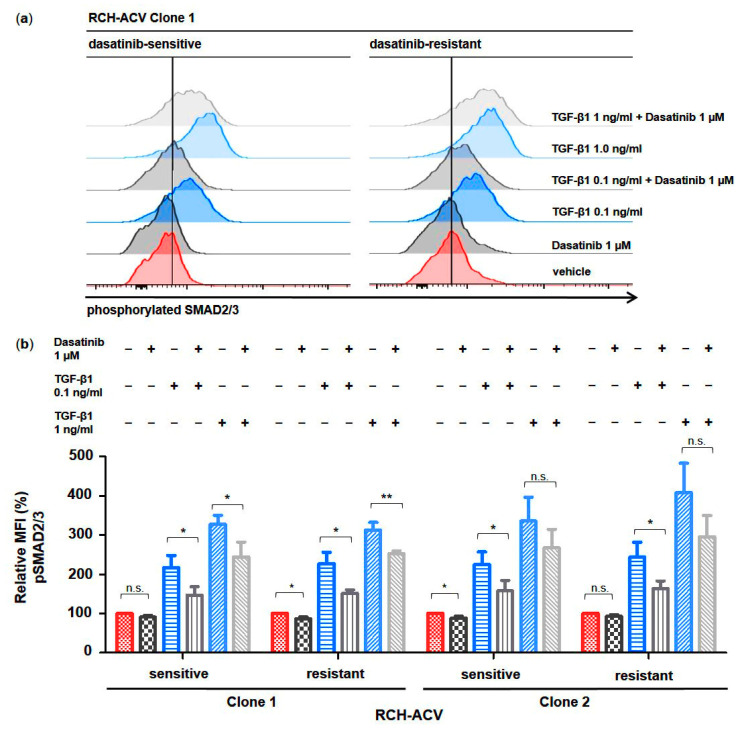
Dasatinib partially blocks TGF-β1-induced phosphorylation of SMAD2/3 in human E2A-PBX1^+^ ALL cells. Phospho-specific intracellular flow cytometry assay in dasatinib-sensitive and -resistant RCH-ACV cells after treatment for 30 min with vehicle, dasatinib or TGF-β1 alone and in combination. (**a**) Histograms of clone 1 (representative of three independent experiments) show phosphorylation state of SMAD2/3 proteins. (**b**) Bar graphs represent median fluorescence intensity (MFI) values expressed as relative to vehicle-treated cells. Statistical analysis by Student’s *t*-test. Data represent the mean of three independent experiments, with error bars indicating ± standard deviation (SD). Differences between mean were considered significant at *: *p* < 0.05; **: *p* < 0.01; n.s.: not significant.

**Figure 3 cancers-15-04328-f003:**
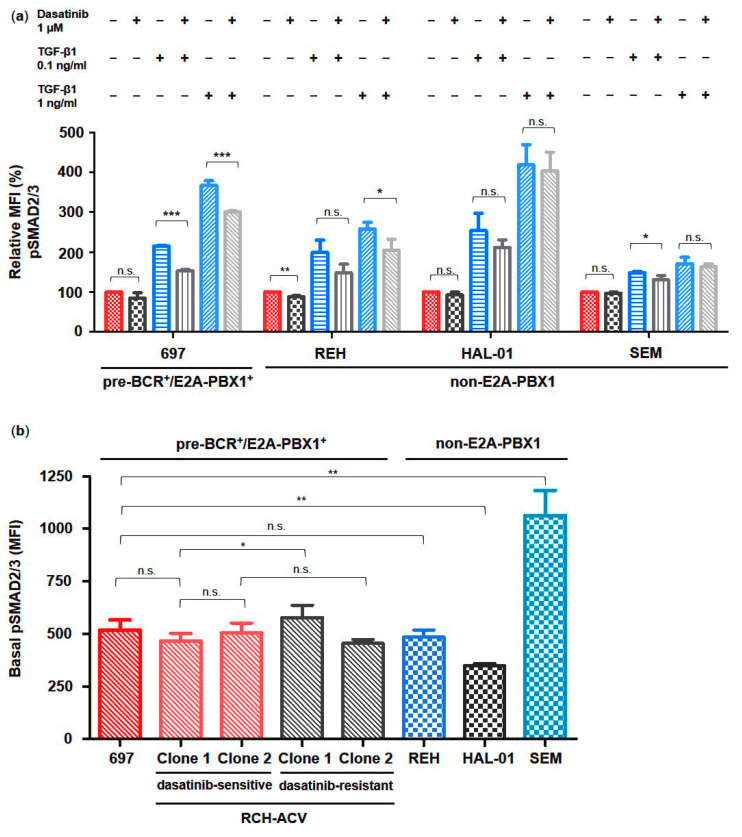
TGF-β signaling activation via phosphorylation of SMAD2/3 is heterogeneous in human BCP-ALL cells. Phospho-specific intracellular flow cytometry assay in dasatinib-sensitive and -resistant RCH-ACV cells after treatment for 30 min with vehicle, dasatinib or TGF-β1 alone and in combination. (**a**) Bar graphs represent median fluorescence intensity (MFI) values expressed as relative to vehicle-treated cells. (**b**) Bar graphs represent basal phosphorylation levels of SMAD2/3 proteins in human pre-BCR^+^/E2A-PBX1^+^ as well as non-E2A-PBX1 BCP-ALL cell lines, which are depicted according to their sensitivity to dasatinib, respectively. Statistical analysis by Student’s *t*-test. Data represent the mean of three independent experiments, with error bars indicating ± standard deviation (SD). Differences between mean were considered significant at *: *p* < 0.05; **: *p* < 0.01; ***: *p* < 0.001; n.s.: not significant.

**Figure 4 cancers-15-04328-f004:**
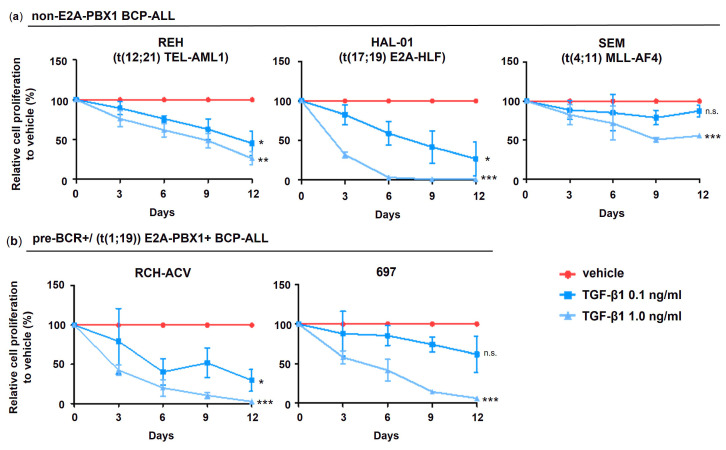
TGF-β1 inhibits cell proliferation in human BCP-ALL cells. Cell number was assessed every 3 days by trypan blue exclusion assay. (**a**) Pre-BCR^+^/E2A-PBX1^+^ and (**b**) non-E2A-PBX1 ALL cells. Cell proliferation is expressed as relative to vehicle-treated cells calculated from accumulative data. Data represent the mean and ±SD of three independent experiments. Statistical analysis by Student’s *t*-test (*: *p* < 0.05; **: *p* < 0.01; ***: *p* < 0.001; n.s.: not significant). Pre-BCR: pre-B cell receptor; B-ALL: B acute lymphoblastic leukemia.

**Figure 5 cancers-15-04328-f005:**
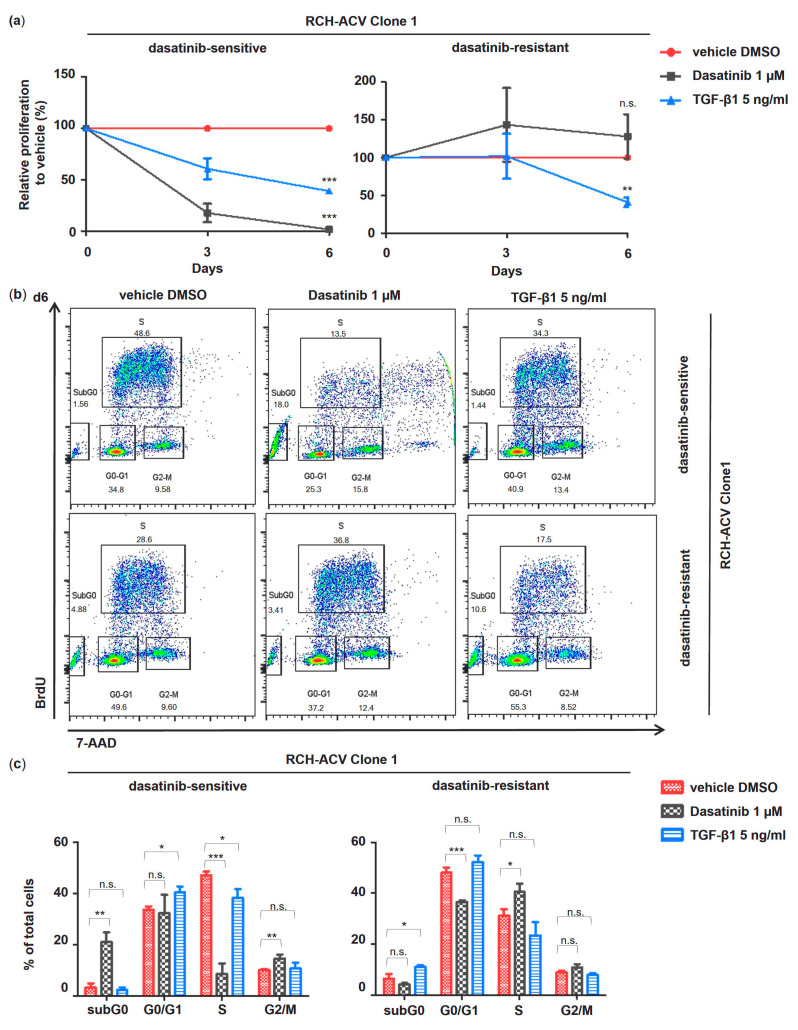
TGF-β1 inhibits cell proliferation in E2A-PBX1^+^ dasatinib-sensitive and -resistant RCH-ACV cells. Comparative pharmacological long-term treatment in E2A-PBX1^+^ dasatinib-sensitive and -resistant RCH-ACV clone 1 over 6 days with vehicle, dasatinib and TGF-β1. The concentration of TGF-β1 was optimized to detect an effect in cell cycle and apoptosis analysis. Cell proliferation was markedly reduced at concentrations of 0.5 and 1.0 ng/mL; however, correlating changes in cell cycle or apoptosis assays were detected at 5.0 and 10.0 ng/mL by flow cytometry. Further experiments were performed with TGF-β1 5.0 ng/mL. (**a**) Cell proliferation is expressed as relative to vehicle-treated cells calculated from accumulative data. (**b**) Flow cytometry plots of cell cycle analysis at day 6 representative of three independent experiments. BrdU: bromodeoxyuridine; 7-AAD: 7-aminoactinomycin D. (**c**) Bar graphs represent quantification of cell cycle analysis at day 6. Data are shown as mean of three independent experiments, with error bars indicating ± SD. Statistical analysis by Student’s *t*-test (*: *p* < 0.05; **: *p* < 0.01; ***: *p* < 0.001; n.s.: not significant).

**Figure 6 cancers-15-04328-f006:**
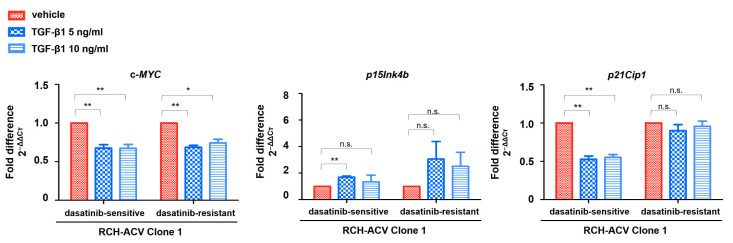
Expression of TGFβ target genes involved in cell cycle arrest suggests differential regulation in dasatinib-sensitive and -resistant RCH-ACV cells. Gene expression analysis following 6 days of treatment with vehicle or TGF-β1. The concentrations of dasatinib and TGF-β1 were optimized to detect an effect on gene expression. Gene expression was analyzed using RT-qPCR. GAPDH was used as housekeeping gene. Mean CT values were normalized to GAPDH and expressed as fold difference in expression (2^−ΔΔCT^) relative to vehicle-treated cells. Bars represent the mean of three independent experiments, with error bars indicating ± SD. Statistical analysis by Student’s *t*-test (*: *p* < 0.05; **: *p* < 0.01; n.s.: not significant).

## Data Availability

The data presented in this study are available on request from the corresponding author.

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
