# Peer review of "Functional Characterization of Transforming Growth Factor-β Signaling in Dasatinib Resistance and Pre-BCR+ Acute Lymphoblastic Leukemia"

_cancers, 2023, doi:10.3390/cancers15174328_

Round 1

Reviewer 1 Report

In the present manuscript, Duque-Afonso and collaborators demonstrate the antiproliferative effect of TGFβ signaling in BCP-ALL cell lines and dasatinib-resistant and sensitive E2A-PBX1+ RCH-ACV clones. SMAD3 levels were found to be upregulated in dasatinib-resistant E2A-PBX1+ RCH-ACV clones. Dasatinib could prevent to a certain extent pSMAD2/3 activation upon TGFβ stimulus, but no differences were found between dasatinib-resistant and sensitive clones. The authors do not clarify whether SMAD3 plays a significant role in TGFβ-induced antiproliferative effects in the context of dasatinib resistance in BCP-ALL.

Major criticisms

1) In figure 1 RNAseq data showed differentially expressed genes associated TGFβ signaling pathway when comparing dasatinib to sensitive E2A-PBX1+ ALL cells. The authors validate the upregulation of SMAD3, a downstream player of TGFβ signaling, in dasatinib resistant RCH-ACV clones. The authors should contextualize better the possible involvement of the other TGFβ signaling pathway related genes found to be as well upregulated in dasatinib resistant RCH-ACV clones.

2) In figure 3 pSMAD2/3 status was assessed in response to TGFβ and dasatinib treatment in human BCP-ALL cell lines. Cell lines were divided by being E2A-PBX1+ or non-E2A-PBX1 but sensitivity to dasatinib should also be evaluated. Non-E2A-PBX1 BCP-ALL cell lines failed to fully replicate dasatinib negative effect on TGFβ induced pSMAD2/3 levels. Why? Are they not sensitive to dasatinib? Or sensitivity to dasatinib does not correlate with the ability of dasatinib to block TFGβ-induced pSMAD2/3 thus confirming previous results with the RCH-ACV clones? If not, SMAD3 total levels could also be determined and compared between E2A-PBX1+ and non-E2A-PBX1 cell lines.

3) The authors should evaluate the functional impact of treating leukemic cells concomitantly with dasatinib and TGFβ.

4) The PI3K/Akt pathway is known to antagonize TGFβ/SMAD3 signaling by inhibiting FOXO localization in the nucleus. It would be of interest to address if inhibition of the PI3K/Akt pathway could paradoxically be pro-proliferative by counteracting TGF-β–induced antiproliferative effect.

Author Response

In the present manuscript, Duque-Afonso and collaborators demonstrate the antiproliferative effect of TGFβ signaling in BCP-ALL cell lines and dasatinib-resistant and sensitive E2A-PBX1+ RCH-ACV clones. SMAD3 levels were found to be upregulated in dasatinib-resistant E2A-PBX1+ RCH-ACV clones. Dasatinib could prevent to a certain extent pSMAD2/3 activation upon TGFβ stimulus, but no differences were found between dasatinib-resistant and sensitive clones. The authors do not clarify whether SMAD3 plays a significant role in TGFβ-induced antiproliferative effects in the context of dasatinib resistance in BCP-ALL.

Major criticisms

  • In figure 1 RNAseq data showed differentially expressed genes associated TGFβ signaling pathway when comparing dasatinib to sensitive E2A-PBX1+ ALL cells. The authors validate the upregulation of SMAD3, a downstream player of TGFβ signaling, in dasatinib resistant RCH-ACV clones. The authors should contextualize better the possible involvement of the other TGFβ signaling pathway related genes found to be as well upregulated in dasatinib resistant RCH-ACV clones.

  • Answer: We thank the reviewer #1 for the comment. RNAseq identified upregulation of several members of the TGFβ superfamily as TGFβ2 and SMAD3 as well as of the bone morphogenetic protein (BMP) subfamily, as BMPR1A, BMP2 and BMP8B, in dasatinib-resistant preBCR+/E2A-PBX1+ ALL cells. Both are involved in regulating a broad spectrum of different functions in adult tissues as well as embryonic development. However, we decided to continue further functional studies investigating the role of the TGFβ/SMAD signaling pathway in our preclinical model of dasatinib-resistance in RCH-ACV cell clones as well as in several BCP-ALL cell lines because of its well-established roles in cell proliferation and cancer. We cannot exclude the involvement of BMP signaling in contributing to the resistant phenotype of RCH-ACV cells to dasatinib, which should be investigated in future studies. We added the figure S1 and comments on page 4, line 207-211 to page 5, line 218-227, and have rewritten the first paragraph of the discussion (page 11, lines 462-470).

  • In figure 3 pSMAD2/3 status was assessed in response to TGFβ and dasatinib treatment in human BCP-ALL cell lines. Cell lines were divided by being E2A-PBX1+ or non-E2A-PBX1 but sensitivity to dasatinib should also be evaluated. Non-E2A-PBX1 BCP-ALL cell lines failed to fully replicate dasatinib negative effect on TGFβ induced pSMAD2/3 levels. Why? Are they not sensitive to dasatinib? Or sensitivity to dasatinib does not correlate with the ability of dasatinib to block TFGβ-induced pSMAD2/3 thus confirming previous results with the RCH-ACV clones? If not, SMAD3 total levels could also be determined and compared between E2A-PBX1+ and non-E2A-PBX1 cell lines.

  • Answer: We thank the Reviewer #1 for this comment. Drug-sensitivity assays with dasatinib using the different BCP-ALL cell lines have been performed previously in Duque-Afonso et al., Cancer Res 2018 and described in this ms. on page 7, line 293-297. As explained by the Reviewer #1, the sensitivity to dasatinib does not correlate with the ability of dasatinib to block TGFb induced pSMAD2/3 or with basal pSMAD2/3. We added a comment on page 12, line 534-535. We agree with the reviewer that SMAD3 total protein assessment to compare between E2A-PBX1+ and non-E2A-PBX1 cell lines could be performed in order to answer the reviewer’s question.

  • The authors should evaluate the functional impact of treating leukemic cells concomitantly with dasatinib and TGFβ.

  • Answer: We showed in this and previous manuscripts that dasatinib inhibits efficiently the cell proliferation of several ALL cell lines with different karyotypes. In this manuscript, we showed that TGFβ treatment impact the cell proliferation of several ALL cell lines through activation of the TGFβ/SMAD pathway. Dasatinib blocks, at least partially, TGFβ/SMAD pathway activation. Therefore, we expect that the concomitant treatment of dasatinib and TGFβ will lead to an antagonistic, but not to an synergistic effect on cell proliferation. We agree with the reviewer that the functional implication of combined treatment with dasatinib and TGFβ in ALL cell lines should be investigated in future studies. A comment was added on page 13, line 553-555.

  • The PI3K/Akt pathway is known to antagonize TGFβ/SMAD3 signaling by inhibiting FOXO localization in the nucleus. It would be of interest to address if inhibition of the PI3K/Akt pathway could paradoxically be pro-proliferative by counteracting TGF-β–induced antiproliferative effect.

Answer: We thank the reviewer #1 for this comment. In our previous studies, we showed the important role of the PI3K/AKT/MTOR pathway in E2A-PBX1+ ALL and we establish combination targeting therapies with promising preclinical efficacies (Grüninger et al., 2022). The goal of our present work was primarily to functionally characterize TGFβ/SMAD3 signaling in leukemogenesis of BCP-ALL and in dasatinib-resistance of preBCR+/E2A-PBX1+ ALL. However, we agree with the reviewer that PI3K/Akt pathway is a well-known opponent of TGFβ-mediated cell-cycle arrest. It has been previously shown that preBCR+/E2A-PBX1+ ALL cells are dependent on hyperactivated PI3K/AKT signaling, therefore, we would rather hypothesize that inhibition of PI3K/AKT pathway might have additive effects to antiproliferation mediated by TGF-β1. We added a comment on page 13, lines 575-587 and a suggestion for further studies that are needed to evaluate this interesting question

Reviewer 2 Report

The Manuscript “Functional characterization of transforming growth factor-ß2 signaling in dasatinib resistance and pre-BCR+ acute lympho-blastic leukemia” by Mostufi-Zadeh-Haghighi et al, provides the definition of role that TGFß signaling pathway plays in leukemogenesis of BCP-ALL as well as in secondary drug resistance to dasatinib. The paper is interesting from a clinical point of view, since there is a continuous needing for the development of novel treatment approaches, but it is possible to individuate major and minor issues.

Major

1)      More than 2 resistant clones shoud be used

2)      236-244 lines should be rewritten in order to better delineate the concepts

Minor

1)      Figures more definite

2)       Cytometric hystogram eith better resolution

3)       figure 3 should be reformulated for the comparison among BCP ALL cells and Dasanitib resistant/sensitive clones

4)      In line 273 the sentence . The concentration of dasatinib and TGF-ß1 were optimized to detect an effect in cell cycle and apoptosis analysis “ should be followed by the optimization process description

The English language appears fluently enough to be followed by the readers in almast all lines of the Manuscript

Author Response

The Manuscript “Functional characterization of transforming growth factor-ß2 signaling in dasatinib resistance and pre-BCR+ acute lympho-blastic leukemia” by Mostufi-Zadeh-Haghighi et al, provides the definition of role that TGFß signaling pathway plays in leukemogenesis of BCP-ALL as well as in secondary drug resistance to dasatinib. The paper is interesting from a clinical point of view, since there is a continuous needing for the development of novel treatment approaches, but it is possible to individuate major and minor issues.

Major

  • More than 2 resistant clones should be used

  • Answer: We apologize to reviewer #2 that we are not able to perform the studies in more than two dasatinib-resistant cell clones. The preclinical model of dasatinib-resistance in preBCR+/E2A-PBX1+ ALL was previously established with two sublines of RCH-ACV cells (Duque-Afonso et al., 2018) and generated through multiple passages over several months. All previous work of our group has been performed using these two subclones of dasatinb-resistant RCH-ACV cells.

  • 236-244 lines should be rewritten in order to better delineate the concepts

  • Answer: As suggested by the reviewer, we have rewritten and reorganized the corresponding paragraph, lines 321-326.

Minor

  • Figures more definite

  • Answer: We thank the reviewer #2 for this suggestion, we optimized the quality of the figures.

  • Cytometric hystogram eith better resolution

  • Answer: As suggested by the reviewer, we optimized the resolution of the cytometric histogram in figure 2.

  • figure 3 should be reformulated for the comparison among BCP ALL cells and Dasanitib resistant/sensitive clones

  • Answer: We thank the reviewer #2 for this suggestion. We reformulated figure 3 and the corrresponding section of the figure legend for better comparison among BCP-ALL cell lines and dasanitib-sensitive and - resistant clones.

  • In line 273 the sentence . The concentration of dasatinib and TGF-ß1 were optimized to detect an effect in cell cycle and apoptosis analysis “should be followed by the optimization process description

  • Answer: We thank the reviewer #2 for this comment. We now added a comment describing the optimization process in the methods section (page 4, line 175-180) as well as in the figure legends of Fig. 5, S3, S4.

Round 2

Reviewer 1 Report

While not performing requested experiments where it was the case, the authors replied reasonably well to the critiques and have altered the manuscript accordingly.